# Making Body Movement in Sign Language Corpus Accessible for Linguists and Machines with Three-Dimensional Normalization of MediaPipe

**Victor Skobov[1]** and **Mayumi Bono[2]**

[1]The Graduate University for Advanced Studies (SOKENDAI), Kanagawa, Japan

[12]National Institute of Informatics, Tokyo, Japan

{[1]vskobov, [2]bono}@nii.ac.jp

## Abstract

Linguists can access movement in the sign language video corpus through manual annotation or computational methods. The first relies on a predefinition of features, and the second requires technical knowledge. Methods like MediaPipe and OpenPose are now more often used in sign language processing. MediaPipe detects a two-dimensional (2D) body pose in a single image with a limited approximation of the depth coordinate. Such 2D projection of a three-dimensional (3D) body pose limits the potential application of the resulting models outside the capturing camera settings and position. 2D pose data does not provide linguists with direct and human-readable access to the collected movement data. We propose our four main contributions: A novel 3D normalization method for MediaPipe's 2D pose, a novel human-readable way of representing the 3D normalized pose data, an analysis of Japanese Sign Language (JSL) sociolinguistic features using the proposed techniques, where we show how an individual signer can be identified based on unique personal movement patterns suggesting a potential threat to anonymity. Our method outperforms the common 2D normalization on a small, diverse JSL dataset. We demonstrate its benefit for deep-learning approaches by significantly outperforming the pose-based state-of-the-art models on the open sign language recognition benchmark.

## 1 Introduction

Our research aims to find a movement representation that allows processing of sign language movement directly, without relying on annotations or systems of predefined movement features, such as the Hamburg Notation System (Prillwitz et al., 1989). And help to overcome the camera settings constraints of the available datasets.

### 1.1 Problem

Due to its visual nature, sign language data are stored and distributed in a video format. Lin-guists must annotate the sign language features on the video to process them. And to annotate a feature, it must first be clearly defined and thoroughly explained to the annotators. That process is not only time-consuming, but it also limits access to the collected data and the processing potential. Pose estimation methods like OpenPose (Cao et al., 2017, 2021) and MediaPipe (Lugaresi et al., 2019) are more often included in the sign language processing pipeline. Recently published sign language datasets often include pose estimation data. How2Sign[1], a large multimodal dataset of American Sign Language (ASL) presented in Duarte et al. (2021), and the Word-Level American Sign Language[2] (WLASL) video dataset presented in Li et al. (2020a), both provide estimated pose data. However, the detection accuracy of pose estimation techniques still requires improvement. Moryossef et al. (2021) reported the negative influence of inaccurate or missing estimations on model performance and applicability beyond training datasets.

Generally, sign language researchers independently develop their own ways of processing pose data for specific body joints and features. Recent approaches still rely on raw pixel data (Sadeghzadeh and Islam, 2022) or a combination of pixel and pose data (Shi et al., 2021). Moreover, sign language datasets vary in terms of the position of the camera relative to the signer, resulting in dissimilarity in 2D pose estimation output for similar movements. Such inconsistencies prevent model generalization, thereby limiting movement feature extraction and inference outside the dataset.

The commonly used standard normalization process proposed in Celebi et al. (2013), recently adopted in Schneider et al. (2019) and Fragkiadakis et al. (2020), involves coordinates axis origin translation to a "neck key point" and scaling of all coordinates so that the distance between shoulder

---

[1]Available at: https://how2sign.github.io/

[2]Available at: https://dxli94.github.io/WLASL/

key points equals one. This work will refer to this normalization method as the "*basic*." This method successfully eliminates the influence of body size differences. However, features such as body rotation (toward the camera), posture, and body proportions in the dataset can still influence feature extraction.

## 1.2 Related Work

Overcoming the camera setting boundaries and pose estimation are being actively researched. Complete 3D body pose estimation using a single camera is one of the main goals (Ji et al., 2020). Activity classification results obtained using 2D and 3D pose data were compared, with no significant difference emerging; the features in 2D data were sufficient (Marshall et al., 2019). A pipeline that includes learning 3D pose coordinates from 2D pose data collected from sign language videos was proposed and used recognizing tasks and synthesizing avatar animations (Brock et al., 2020). The Skeletor (Jiang et al., 2021), a deep-learning approach that refines the pose's estimated coordinates in three dimensions. However, it relies on contextual information and therefore requires a sequence of frames.

## 1.3 Our Proposal

Here, we propose a three-dimensional normalization and encoding method for MediaPipe pose data that is entirely algorithmic and normalizes pose data based on fixed human proportions. Data is presented as a series of angular joint changes. It could be applied to various sign language processing tasks and datasets and allows the search and extraction of movement features. Both methods are detailed in Sections 2.1 and 2.3.

We tested our method against the basic normalization on continuous sign language data using standard machine learning techniques in Section 3.1 and isolated sign language data using the deep-learning in Section 3.2. We show how the JSL sociolinguistic features are present in the signing movement and how they can be explored using our methods.

The main contributions are:

- Novel three-dimensional normalization method for MediaPipe

- Novel movement representation method for a direct sign language processing

- An analysis of JSL sociolinguistic features present in movement patterns

- Substantially outperforming state-of-the-art results on the WLASL-100 public benchmark for pose-based deep-learning methods

Our solution and reproduction code are available as a *mp2signal* python package and as a GitHub repository[3].

## 2 Methodology

To achieve our goal of directly processing sign language, we set the following requirements for the desired movement representation: the adherence to *the triangle inequality*, the capability of movement synthesis, being intuitive and understandable to humans, and being low-dimensional.

The adherence triangle inequality is essential for automated data processing techniques like clustering, machine learning, and indexing.

Movement representation data must be distributed across a space compatible with the notion of distance and similarity. Sampling from such a space should return the corresponding pose, and moving through it should produce movement to meet the movement synthesis requirement.

The space should not have any latent features, and its dimensions must be perceivable by a human. To promote readability and facilitate processing, the space must be as low-dimensional as possible to eliminate unnecessary information from representations.

Normalization must transform pose data into the desired space, and encoding must represent it suitably for human perception.

To determine the degree of adherence to these requirements and compare our method to the basic normalization method, mentioned in Section 1.1, we conducted experiments on two types of sign language data using standard machine learning and deep-learning techniques.

## 2.1 3D Normalization

The MediaPipe's holistic model[4] along with two $x$ and $y$ provides a limited estimation of the $z$ coordinate for each body key point on the given frame. We propose a procedure to improve the depth $z$ coordinate estimated by MediaPipe.

---

[3]Available at: https://github.com/vskobov/mp2signal

[4]Available at: https://google.github.io/mediapipe/solutions/holistic.html

Joints do not move in isolation; they naturally interact within the skeleton. For example, the movement of the arm changes the position of the hand and its fingers. Therefore, we propose processing pose skeleton data as a *"joint tree structure"* that respects actual human proportions, with the root node in the neck key point. We aimed to use all available information to simplify pose data processing. We selected 137 joints from the holistic model: 22 for each hand, nine for the upper body, and 84 for facial expression. We created a human skeleton, which showed rigidness, proportions, and connectivity in line with the human body.

To improve the depth coordinates estimated by MediaPipe, we use the proportions of the body. In Herman (2016), an overview of standard human body proportions is provided. For simplification, we assume that the body proportions for all data processed with MediaPipe are constant when using our method.

We captured a short video of a body standing upright and used MediaPipe to calculate the ratio of distances between key points relative to the distance between the shoulders. The maximum distances across frames were used to calculate the proportions. The *joint tree* model of the human body aligned with MediaPipe key points stores the proportional values for each joint. From the length of just one joint in real space, we can compute the lengths of all joints, which requires some reliable MediaPipe estimation as a basis.

The holistic MediaPipe model includes a face model for estimating key facial points; its depth coordinates are the most accurate. The distance between the eyes in the MediaPipe model is selected as a basis for calculating the lengths of body joints. We trust its estimation the most. Eyes positions are calculated based on the average positions of key points *159*, *143*, *157*, and *149* for the left eye, and key points *384*, *386*, *379*, and *372* for the right eye. Relative to the distance between the shoulders, the distance between the eyes was calculated as *0.237* from the previously captured short video.

With this ratio, we calculate the lengths of all body joints in 3D using the Formula 1, where $eyedistance$ gives the distance between the eyes according to MediaPipe and $prop_j$ gives the *"captured proportion"* of the joint. We calculate the $z$ coordinate using the length, and relative $x$ and $y$ coordinate with the origin in the *"parent joint"* in the *joint tree structure*. Lastly, we apply the sign

value from MediaPipe's original $z$ estimation to it.

$$length_j = prop_j * (eyedistance/0.237) \quad (1)$$

The *joint tree structure* allows us to control the order of calculation with the traversal and process only the desired part of the tree if needed. To obtain the coordinates for a joint with the origin set at the neck point, we sum the coordinates for all its parent joints in the tree. A detailed example of the 3D estimation step is shown in the middle part of Figure 1.

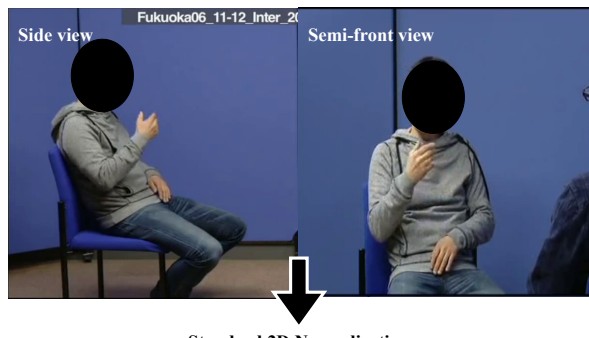

Figure 1: Normalization of a pose captured from the side *(top-left)* and "semi-front" *(top-right)*, basic normalization *(middle)*, our 3D normalization *(down)*.

## 2.2 Scaling and Rotation

After 3D coordinates refinement, the pose data are represented as a *joint tree structure* with coordinates in 3D space. To address the variation in camera angle and relative position, we rotate and scale the coordinates as the final part of the normalization process. The resulting pose is consistently rotated toward the camera and is fixed in size. The

root node (neck point) is the origin *(0,0,0)*, and the left shoulder is at *(0.5,0,0)*.

Both scaling and rotation are performed through a linear transformation in 3D space. To generate the transformation matrix, a scaling factor and rotation angles are required, which we compute for each frame. We apply the transformation matrix to all joint coordinates, using *joint tree structure* traversal to obtain rotated and scaled coordinates for each joint. A detailed example of a scaled and rotated pose is shown in Figure 1 *(bottom panel)*.

Facial expressions are essential for processing sign language. Therefore, we perform an additional separate transformation only for face points. We scale and rotate the key face points so that the nose points toward the camera along the *z*-axis, while the point between the eyes is on the *y*-axis. Additional normalized facial key point data are shown in Figure 1 *(bottom panel, upper right corners)* and 2e–h.

## 2.3    Representation

Our normalization process returns pose data as 137 "tree-structured joints" with 3D coordinates, which is helpful for decomposing movement. We use *relative coordinates* for each joint, with the origin set at the parent joint, to represent the joint's movement in space independently. Since the proportions are fixed and known, independent movement may be estimated with *arccosine* of *direction cosines* values, i.e.,the angles between joint and axes, which range from $-\pi$ to $\pi$ in radians. The resulting body movement appears as a series of 411 $(3*137)$ isolated signals. Each signal shows a value for the angle between the corresponding joint and the corresponding axis at every frame. The resulting decomposition allows the quick determination of when, where, and what movement of the body is captured, providing direct access to it.

Initially, for each key body point, we obtained three values from the MediaPipe. After normalization, the key body points became joints with three *direction angles* values, striped from the variation in size, rotation, and body proportions. The dimensionality of the information remained the same while the representation space changed, adhering to the requirements in Section 2.

We use image RGB color space to visualize a series of direction angles for joints to simplify the interpretation of the movement. The process is shown in Figure 2a–b: direction angles with *x*-, *y*-,

and *z*-axes ranging from $-\pi$ to $\pi$ in radians are encoded in the red, green, and blue channels, respectively, as 8-bit integers ranging from 0 to 255. Figure 3 shows an example image of a representative movement.

For handshapes, it might serve as an additional visual clue to add one-dimensional (1D) encoding of the absolute 3D angle (0°–180°; blue = 0°, red = 90°, green = 180°) between the hand joints and their parents. Figure 2d provides an example.

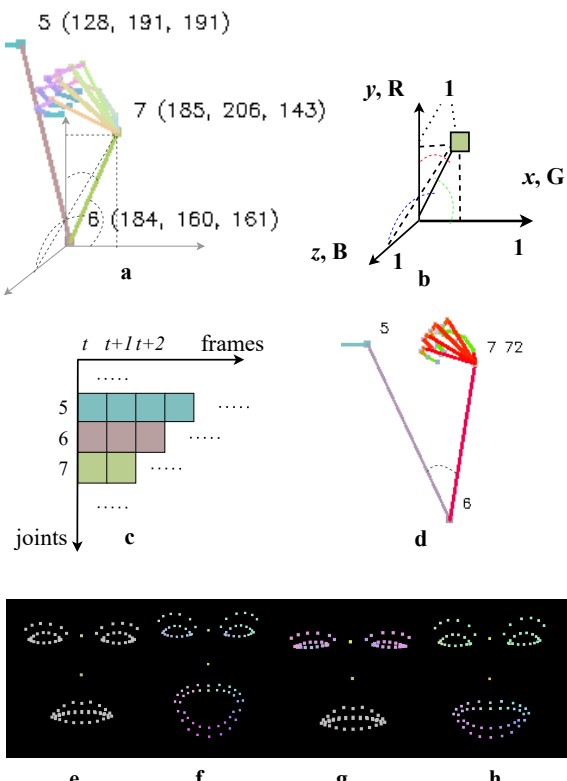

Figure 2: Proposed encoding scheme: a) calculating a joint's direction angles with *arccosine*; b) translating the angle values into RGB color space; c) displaying the RGB-encoded data as an image; d) adding additional 3D angles between the joints with color-coding; e) normalizing facial expressions with a relaxed baseline face; and f–g) example color-coding of various facial expressions deviating from the relaxed baseline face.

Key face points are encoded differently. We captured a relaxed facial expression as the baseline (Figure 2e) and encoded deviation angles for each key point in three axes. The angle changes are usually tiny, so we multiplied the difference by a factor of four (determined by trial and error) to boost visual interpretation. Figure 2f–g shows an example of encoded facial expressions.

For computer processing methods, channels must be separated; thus, data will be encoded as an

8-bit integer with only one channel per pixel. This method allows for movement manipulation via image manipulation. Image processing is a highly advanced area with various methods and approaches applicable to movement processing. Movement detection and recognition can borrow approaches from object detection (image processing).

Figure 4 shows an encoded representation of 13 simultaneous closings and openings of a rotating fists. Where and when the movement occurs is easily detectable with the naked eye, and might also be easily detectable via modern pattern-recognition methods.

The proposed encoding scheme is straightforward, and the image can easily be decoded back to direction angles and coordinates. Movement patterns are fully explainable and can produce skeleton animations, aiding visual comprehension and thus satisfying the requirements in Section 2.

We hope to encourage researchers to explore the capabilities of encoded movement data, augmenting their sign language knowledge to explore movement features. Section 3 discusses how the proposed methods compare to the standard normalization process used for linguistic and sociolinguistic features extracted from sign language datasets.

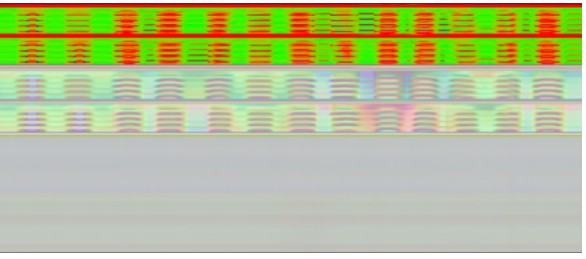

Figure 4: Hands movement is clearly visible on the encoding plot of a processed video sample with 13 closings and openings of rotating fists.

this Section, we compare the performance of our method to the basic normalization method, mentioned in Section 1.1, on a dataset composed of continuous samples — the JSL Dataset and a public benchmark dataset collected of isolated samples — the WLASL-100 dataset. The JSL dataset has a variation in the camera angle and includes coding for various sociolinguistic features. However, it is a small and very diverse dataset; therefore, it will be used for feature exploration and camera settings boundary testing using standard machine learning algorithms. The isolated WLASL-100 is more suitable for deep-learning testing since it is an established public sign language recognition benchmark.

### 3.1 Continious Signs - JSL

We created a simple dataset from the Japanese Sign Language (JSL) Corpus presented by Bono et al. (2014, 2020). The JSL Corpus is a continuous and dialog-based corpus that includes utterance- and word-level annotations. It consists of conversations between two participants freely discussing various topics. The signers vary in age, geographic area, gender, and the deaf school they attended. Conversations were captured from both the semi-front and side positions; a sample from the dataset is shown in Figure 1.

### 3.1.1 JSL Dataset Satistics

Using the word-level annotations, we have selected lexical signs from the JSL Corpus with more than 25 samples. For each lexical sign, we extracted 25 video examples for each camera view (a total of 50 samples). Some samples had an insufficient capturing quality for pose estimation, so our final dataset comprised 674 semi-front view and 608 side view videos. The resulting number of classes and samples per class for each feature is shown in Table 4.

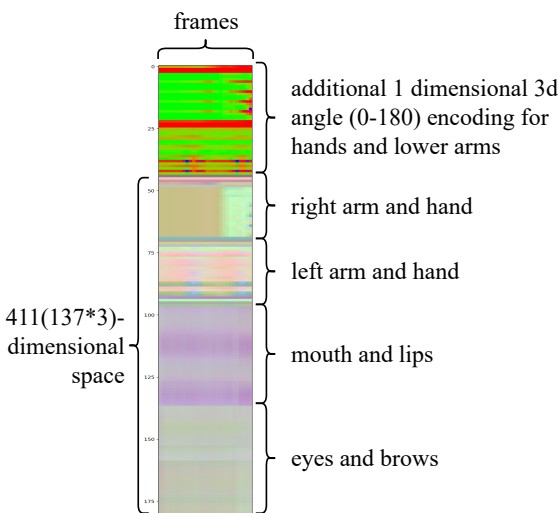

Figure 3: Structure our movement encoding method.

## 3 Experimental Setup

The proposed normalization was explicitly developed to process data with high variance, which is typical of data captured in real life. The decomposition property of our approach allows for comparing pure movement data on a joint-by-joint basis. In

For comparison, we created a second dataset from the same samples by normalizing the MediaPipe pose data using the basic normalization method. The resulting samples vary in duration from four to 120 frames, and we had to resize them using linear interpolation to fit the longest sample in the dataset. The JSL Corpus includes the signer ID, prefecture of residence, deaf school, age group, and gender. This information was added to the dataset since we were interested in examining whether these features affected signing movements.

### 3.1.2 Classification

First, we used the "Neighbourhood Components Analysis" (NCA) approach presented in Goldberger et al. (2004) to visualize the embedding space for each sociolinguistic feature in the dataset. We tested various classification techniques using the *scikit-learn* package [5] (Pedregosa et al., 2011), including the linear support vector classifier (SVC) (Fan et al., 2008), nearest neighbor classifier (Cover and Hart, 1967), naive Bayes (Zhang, 2004), and decision tree (Breiman et al., 1984), to check for the presence of features in the data and assess the potential applicability of our normalization method to classification tasks.

We designed an experiment in which a model was trained on data captured from the front perspective and tested using data captured from the side perspective. We did this to address the camera angle boundary and generalization issue mentioned in Section 1, i.e., to determine the applicability to other datasets and capture conditions.

### 3.2 Isolated Signs - WLASL

We used the popular public deep-learning benchmark, the Word Level American Sign Language Dataset (Li et al., 2020a), to demonstrate the utility of our normalization and representation methods in deep-learning pose-based approaches.

### 3.2.1 WLASL-100 Dataset Satistics

We selected the WLASL-100, a WLASL subset of the top one hundred samples per sign. The split statistics are shown in Table 1.

| Classes | Train | Validation | Test |
|---------|-------|------------|------|
| 100 | 1442 | 338 | 258 |

Table 1: WLASL-100 data subset satistics.

[5]Available at: https://scikit-learn.org

### 3.2.2 WLASL Preprocessing

The WLASL dataset is distributed in video format, requiring preprocessing before training. Our preprocessing flow starts with the mediapipe pose data extraction and normalization using the proposed methods and basic normalization to create two datasets for comparison. The next preprocessing steps are visualized in Figure 8 and include finding and cutting to the part where both hands are visible, resizing using the linear interpolation to a fixed 100 frames, and removing the facial and relative joint information rows from the samples to reduce the dimensionality from 455 and 411 in basic normalization case to 159 values per 100 frames. We want to point out that start and end frames for basic normalization samples were determined using a corresponding sample of the proposed normalization dataset to guarantee consistency between the two datasets.

### 3.2.3 Model

We chose the Conformer model presented by Gulati et al. (2020) as the core unit since it is aimed at two-dimensional signal representations. Figure 5 shows the overview of our model, where we use the adaptive average pool layer to reduce each sample to 15 frames and add one fully connected layer before and one after the conformer. Both do not have bias nor an activation but have an L2 normalization and a dropout layer after. The resulting model ends up as simple as possible. We train it using the Adam optimizer (Kingma and Ba, 2015) and the log loss for 200 epochs with the mini-batch size 32. Before training, all samples are standard-scaled[6] on the training set, and during training, a 50% uniform noise is added to the samples.

## 4 Results and Analysis

### 4.1 Continious Signs

Figure 6 shows the well-distinguished clusters for *Signer ID*, *Prefecture*, *Deaf School*, and *Age Group*, with the only exception for the *Gender* feature. Table 4 shows all classification results and samples per class distribution, whereas Table 2 shows only the best results as a summary.

For the *Lexical Sign* feature, our method outperforms the basic normalization method. *Signer ID* was the best-performing feature on front view data (accuracy = 78.57%) when using naive Bayes, for

[6]Implementation used: https://gist.github.com/farahmand-m/8a416f33a27d73a149f92ce4708beb40

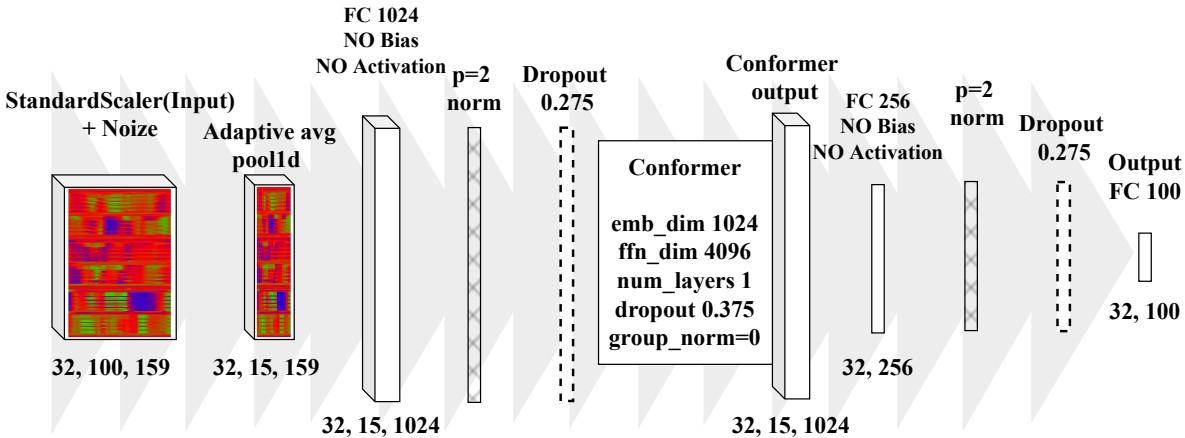

Figure 5: An adaptive average pooling layer, with a single conformer layer (Gulati et al., 2020) trained with cross-entropy loss.

which the baseline was 8.33% and 43/12 samples per number classes ratio. Also, *Lexical Sign* with 49.78% accuracy on front plus side view data from naive Bayes compared to a 3.7% chance of guessing with only 43/27 samples per number classes ratio. Each signer has a unique movement pattern (i.e., signing style). Likewise, stylistic characteristics uniquely vary depending on the prefecture of residence, deaf school, and age group; only gender had no influence on signing movements.

The results, shown in Table 2, indicate that the only feature that did not significantly improve prediction performance from the chance level was *Gender* (accuracy = 75.53%), even though it had the most samples per class among all features and is binary in data. This is consistent with the NCA embedding visualization. We cannot predict the JSL signer's gender based on their signing movements. Model performances on other features were consistently above the baseline, except for the learning transfer tests (training on front view images and testing on side view images), thus confirming the presence of the movement patterns attributed to them. JSL experts validated the results, confirming our findings based on their experience.

The last two columns in Table 2 indicate that our method retains the extracted features better than the basic normalization method for all features, overcoming the camera angle setting boundary.

### 4.2 Isolated Signs

In Table 3, we report the average accuracy across ten runs for each dataset with the top 1, top 5, and top 10 prediction scores as established in the WLASL benchmark reporting practice. Our model

outperforms the state-of-the-art pose-based results on both datasets. Moreover, the proposed normalization pose-only dataset exceeds the models with combined modalities. As for comparing the normalization of two datasets, the results suggest a great performance improvement using the proposed normalization over the basic normalization, going from 75.85% to 84.26% using the same pose estimated data from MediaPipe. In Figure 7, the accuracy curve of the test data set during training is shown, indicating a clear improvement in learning with the proposed normalization.

## 5   Discussion and Conclusions

The proposed methods allow linguists and engineers to directly access the movement captured in the sign language corpus. Before, they had to use human annotation or recognition methods, which both relied on a predefinition of the features and were effectively limited by it.

Sign language movement can now be represented and stored in human-readable form with the proposed encoding method, allowing researchers to observe and comprehend it visually. Normalized pose data are distributed over a joint-based, low-dimensional feature space with distinct and fully explainable dimensions. Machine learning methods can also process it directly since it complies with the distance notion and the triangular inequality.

The embedding results showed the presence of stylistic movement features that correspond to known sociolinguistic features of JSL, similar to predictions of the speaker's country of origin based on their accent. Linguists and sign language ex-

| | Train Dataset | Front + Side | | Side View | | Front View | | Front View | |
| | Test Dataset | Front + Side | | Side View | | Front View | | Side View | |
| Feature | Baseline | Ours | Basic | Ours | Basic | Ours | Basic | Ours | Basic |
|---|---|---|---|---|---|---|---|---|---|
| Lexical Sign | 3.7 | **49.78** | 25.32 | **37.27** | 16.36 | **39.02** | 21.14 | **32.73** | 8.18 |
| Signer ID | 8.3 | **72.64** | 66.04 | 72 | **83.33** | **78.57** | 62.07 | **40** | 25 |
| Prefecture | 25 | 50 | **57.35** | **56.25** | 45.45 | 56.76 | **66.67** | **34.38** | 30.3 |
| Deaf School | 16.67 | **48.05** | 46.75 | **54.05** | 45.95 | **55** | 43.9 | **35.13** | 29.73 |
| Age Group | 16.67 | **49.12** | 42.1 | 44.44 | **51.72** | **45.16** | 39.29 | **33.33** | 24.14 |
| Gender | 50 | **76.88** | 70.52 | 71.61 | **76.83** | 76.34 | **78.02** | **75.31** | 57.32 |

Table 2: Comparison of the normalization methods in terms of JSL feature classification performance.

| Model | Pose-based | Frame+ Backbone | Top 1 | Top 5 | Top 10 |
|---|---|---|---|---|---|
| Li et al. (2020a) | | ✓ | 65.89 | 84.11 | 89.92 |
| Li et al. (2020b) | | ✓ | 77.55 | 91.42 | - |
| Hosain et al. (2021) | | ✓ | 75.67 | 86.42 | 90.16 |
| Maruyama et al. (2021) | ✓ | ✓ | **81.38** | 94.13 | 96.05 |
| Tunga et al. (2021) | ✓ | | 60.15 | 83.98 | 88.67 |
| Boháček and Hrúz (2022) | ✓ | | 63.18 | - | - |
| Maruyama et al. (2021) | ✓ | | 71.07 | 90.13 | 92.42 |
| Naz et al. (2023) | ✓ | | 72.09 | 88.76 | 92.64 |
| Ours (basic norm) Agv, Std | ✓ | | **75.85**($\pm$1.14) | **92.75**($\pm$0.6) | **95.27**($\pm$0.48) |
| Ours (basic norm) Max | ✓ | | 77.51 | 93.8 | 96.12 |
| Ours (3D norm) Avg, Std | ✓ | | **84.26**($\pm$0.82) | **95.66**($\pm$0.42) | **96.86**($\pm$0.27) |
| Ours (3D norm) Max | ✓ | | 85.27 | 96.12 | 97.28 |

Table 3: The comparison of accuracy scores on the WLASL-100 test data. We report the performance of our model with the proposed and basic normalization method.

perts can apply their knowledge of language properties and the proposed method to uncover novel features. Nevertheless, our results raise a concern about signer privacy protection since stylistic features of individual signers can be predicted based solely on signing movement.

The deep-learning WLASL-100 benchmark results are consistent with the JSL dataset tests. Our method significantly outperforms other pose-based methods and successfully competes with multimodal approaches. Sign language is naturally conveyed through body movement; extracting it from the collected video data improves performance and robustness.

Our method performs consistently well across all data sets. We satisfied the initial requirements outlined in Section 2 and addressed the generalization issue discussed in Section 1. The proposed methods are suitable for any sign language, and multiple sign languages can be encoded into one signing space, thus facilitating cross-language studies in future research.

## Limitations

The proposed representation method can be used for any three-dimensional pose estimation. However, the proposed normalization method relies entirely on the initial data recording quality and estimation accuracy of MediaPipe and is incompatible with two-dimensional pose estimation methods like OpenPose. Our normalization method recalculates the value of $z$ coordinate but relies on MediaPipe's depth estimations to determine the order of the final coordinates. Even under ideal conditions, accounting for body proportions is difficult since the normalization method assumes all humans have the same body proportions. It may lead to instances where the hands are not ideally touching, failing to detect an important sign language feature.

Processing some facial expressions, mouth gestures, and mouthing is limited and requires additional modalities (e.g., pixel data). Still, the detected facial key points can provide aid in pixel extraction.

## Ethics Statement

In our work, we used anonymized pose skeleton data extracted from the published Japanese Sign Language Corpus (Bono et al., 2014, 2020) in accordance with the ACL Ethics Policy. However, one of the work's conclusions proposes that a person can be identified, to a limited degree, using their signing style. We encourage the research community to pay closer attention to the possible development of future authentification systems using movement habits.

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

# A   Appendix

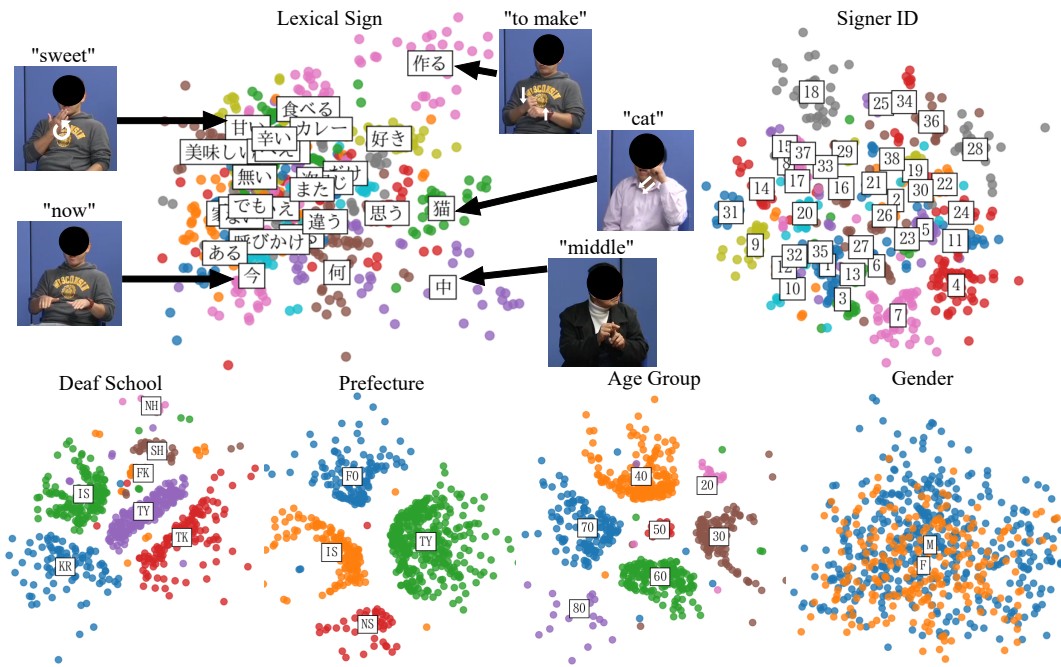

Figure 6: The NCA embedding visualization of various Japanese sign language corpus features.

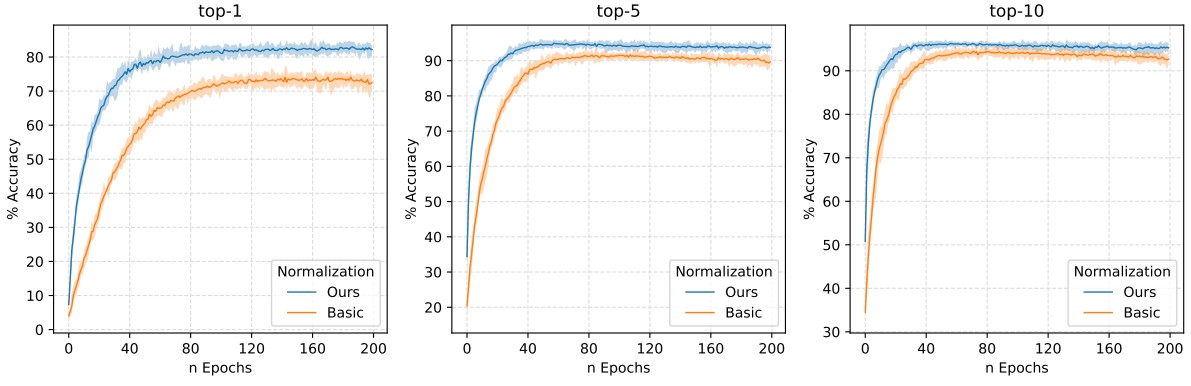

Figure 7: Accuracy *(%)* on top 1 *(left)*, top 5 *(middle)*, and top 10 *(right)* on the test dataset of WLASL-100 dataset during training.

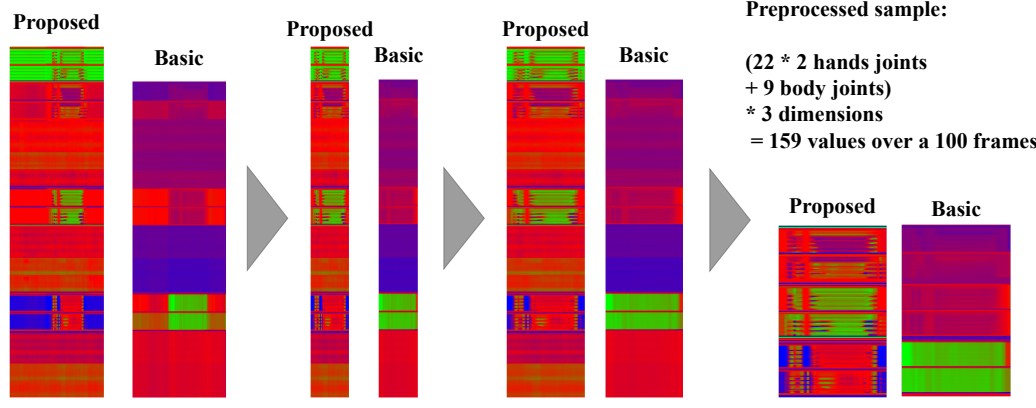

Figure 8: WLASL preprocessing: to simplify the data, we focus only on the parts of the sample where hands are visible. Then, we resize the data to 100 frames and eliminate facial and relative joint information to reduce the complexity.

| Train Dataset | Front + Side | | Side View | | Front View | | Front View | |
|---|---|---|---|---|---|---|---|---|
| Test Dataset | Front + Side | | Side View | | Front View | | Side View | |
| | Ours | Basic | Ours | Basic | Ours | Basic | Ours | Basic |
| **Lexical Sign** | | | | | | | | |
| Baseline: **3.704** | | | Number of Classes: **27** | | Samples per Class: **43** | | | |
| Nearest Neighbors | 30.9 | 12.02 | 27.27 | 15.46 | 28.45 | 12.2 | 27.27 | 5.46 |
| Linear SVM | 49.78 | 25.32 | 37.27 | 14.54 | 39.02 | 21.14 | 32.73 | 3.64 |
| Decision Tree | 19.74 | 17.17 | 25.45 | 16.36 | 23.58 | 20.32 | 18.18 | 8.18 |
| Naive Bayes | 24.03 | 11.16 | 19.09 | 12.73 | 29.27 | 12.2 | 16.36 | 4.54 |
| Best | **49.78** | 25.32 | **37.27** | 16.36 | **39.02** | 21.14 | **32.73** | 8.18 |
| **Signer ID** | | | | | | | | |
| Baseline: **8.333** | | | Number of Classes: **12** | | Samples per Class: **43** | | | |
| Nearest Neighbors | 41.51 | 45.28 | 50 | 39.58 | 37.5 | 36.21 | 40 | 8.33 |
| Linear SVM | 72.64 | 66.04 | 70 | 56.25 | 64.29 | 50 | 34 | 14.58 |
| Decision Tree | 50.94 | 56.6 | 62 | 54.17 | 50 | 62.07 | 20 | 25 |
| Naive Bayes | 66.98 | 39.62 | 72 | 83.33 | 78.57 | 51.72 | 14 | 16.67 |
| Best | **72.64** | 66.04 | 72 | **83.33** | **78.57** | 62.07 | **40** | 25 |
| **Prefecture** | | | | | | | | |
| Baseline: **25** | | | Number of Classes: **4** | | Samples per Class: **84** | | | |
| Nearest Neighbors | 32.35 | 57.35 | 50 | 45.45 | 37.84 | 41.67 | 21.88 | 21.21 |
| Linear SVM | 50 | 39.71 | 50 | 27.27 | 51.35 | 66.67 | 21.88 | 30.3 |
| Decision Tree | 30.88 | 39.71 | 56.25 | 33.33 | 32.43 | 55.56 | 15.62 | 15.15 |
| Naive Bayes | 38.23 | 33.82 | 50 | 33.33 | 56.76 | 63.89 | 34.38 | 21.21 |
| Best | 50 | **57.35** | **56.25** | 45.45 | 56.76 | **66.67** | **34.38** | 30.3 |
| **Deaf School** | | | | | | | | |
| Baseline: **16.667** | | | Number of Classes: **6** | | Samples per Class: **65** | | | |
| Nearest Neighbors | 33.77 | 46.75 | 40.54 | 43.24 | 40 | 41.46 | 27.03 | 18.92 |
| Linear SVM | 42.86 | 46.75 | 48.65 | 45.95 | 42.5 | 43.9 | 29.73 | 29.73 |
| Decision Tree | 28.57 | 32.47 | 40.54 | 32.43 | 37.5 | 31.71 | 35.13 | 21.62 |
| Naive Bayes | 48.05 | 36.36 | 54.05 | 40.54 | 55 | 29.27 | 21.62 | 18.92 |
| Best | **48.05** | 46.75 | **54.05** | 45.95 | **55** | 43.9 | **35.13** | 29.73 |
| **Age Group** | | | | | | | | |
| Baseline: **16.667** | | | Number of Classes: **6** | | Samples per Class: **47** | | | |
| Nearest Neighbors | 31.58 | 28.07 | 18.52 | 37.93 | 32.26 | 39.29 | 33.33 | 17.24 |
| Linear SVM | 45.61 | 42.1 | 29.63 | 37.93 | 38.71 | 28.57 | 22.22 | 24.14 |
| Decision Tree | 31.58 | 35.09 | 33.33 | 27.59 | 35.48 | 32.14 | 25.93 | 10.35 |
| Naive Bayes | 49.12 | 26.32 | 44.44 | 51.72 | 45.16 | 39.29 | 14.81 | 20.69 |
| Best | **49.12** | 42.1 | 44.44 | **51.72** | **45.16** | 39.29 | **33.33** | 24.14 |
| **Gender** | | | | | | | | |
| Baseline: **50** | | | Number of Classes: **2** | | Samples per Class: **423** | | | |
| Nearest Neighbors | 76.88 | 70.52 | 71.61 | 62.2 | 70.97 | 60.44 | 61.73 | 50 |
| Linear SVM | 69.36 | 68.79 | 71.61 | 76.83 | 68.82 | 78.02 | 75.31 | 57.32 |
| Decision Tree | 65.32 | 67.05 | 70.37 | 67.07 | 75.27 | 64.83 | 45.68 | 39.02 |
| Naive Bayes | 65.32 | 61.85 | 54.32 | 71.95 | 76.34 | 58.24 | 61.73 | 48.78 |
| Best | **76.88** | 70.52 | 71.61 | **76.83** | 76.34 | **78.02** | **75.31** | 57.32 |

Table 4: Comparison of the performance of the various standard classification methods for JSL feature prediction.