# OpenReview forum: "Making Body Movement in Sign Language Corpus Accessible for Linguists and Machines with Three-Dimensional Normalization of MediaPipe"
_EMNLP/2023/Conference — EMNLP 2023 Findings_

### Official Review · Reviewer_iboj · 2023-08-05

**Soundness:** 4

**Excitement:**

3: Ambivalent: It has merits (e.g., it reports state-of-the-art results, the idea is nice), but there are key weaknesses (e.g., it describes incremental work), and it can significantly benefit from another round of revision. However, I won't object to accepting it if my co-reviewers champion it.

**Paper Topic And Main Contributions:**

Making Body Movement in Sign Language Accessible

In this paper the authors describe a detailed normalization pipeline to convert annotated sign language images into a standard format. They show that learning algorithms working on their format are able to do better on sign language tasks than alternative normalization schemes.

**Reasons To Accept:**

+ addressing a very important problem
+ generally very clearly written description
+ very detailed normalization scheme taking into account 3D body shapes and a variety of measurements
+ good preliminary experimental data suggesting the scheme is beneficial

**Reasons To Reject:**

- the use of a single "standard" body measurement for reference and normalization seems like a weakness and also raises ethical concerns. Learners working on this representation may suffer from poorer accuracy on body types not matching the "chosen standard". This is analogous to for example how facial recognition systems trained on faces of specific types misidentify faces of others leading to significant harms. I strongly  urge the authors to consider moving away from a single standard for normalization. This is not discussed as a concern by the authors in their ethics section.
- some steps are not adequately described, such as how scaling and rotation are done.
- the figures are not helpful to illustrate the operations. Only the end result is shown not what is done to get there.
- it was not clear to me how z coordinates were being calculated.

**Reproducibility:**

4: Could mostly reproduce the results, but there may be some variation because of sample variance or minor variations in their interpretation of the protocol or method.

**Reviewer Confidence:**

3: Pretty sure, but there's a chance I missed something. Although I have a good feel for this area in general, I did not carefully check the paper's details, e.g., the math, experimental design, or novelty.

---

> ### Author Rebuttal · Authors · 2023-08-28
>
> Comments on Reasons To Reject:
>
>     the use of a single "standard" body measurement for reference and normalization seems like a weakness and also raises ethical concerns. Learners working on this representation may suffer from poorer accuracy on body types not matching the "chosen standard". This is analogous to for example how facial recognition systems trained on faces of specific types misidentify faces of others leading to significant harms. I strongly urge the authors to consider moving away from a single standard for normalization. This is not discussed as a concern by the authors in their ethics section.
>
> Comment:
>
> Thank you. We have been considering addressing it. However, we did not have a proper scientific background to do so correctly and effectively since this is not our field. The following work may be used as guidance on the topic: Herman, I.P. (2016). Terminology, the Standard Human, and Scaling. In: Physics of the Human Body. Biological and Medical Physics, Biomedical Engineering. Springer, Cham. https://doi.org/10.1007/978-3-319-23932-3_1, https://link.springer.com/chapter/10.1007/978-3-319-23932-3_1
>
> The body proportions can be recalculated, of course. If you look at the code, we use it as a dictionary, which can be adjusted.
>
> This is a thoughtful comment; however, is it a reason to reject the work entirely?
>
>     some steps are not adequately described, such as how scaling and rotation are done.
>
> Comment:
>
> We have a dedicated Section 2.2 "Scaling and Rotation" where we describe in detail how the scaling and rotation are done. We use linear transformation (a basic, widely known and used technique in any 3D operations, some course, not ours: https://cseweb.ucsd.edu/classes/wi18/cse167-a/lec3.pdf)
>
>     the figures are not helpful to illustrate the operations. Only the end result is shown not what is done to get there.
>
>     it was not clear to me how z coordinates were being calculated.
>
> Comment:
>
> line 215: we calculate the lengths of all body joints in 3D using the Formula 1, where...
>
> Lenght - is the length in 3D space, that we approximate using the body proportions (Formula 1 , line 224).
>
> line 219: We calculate the z coordinate using the length, and relative x and y coordinate with the origin in the "parent joint"...
>
> Using the ''Pythagorean theorem" :
> length^2 = x^2 + y^2 + z^2 -> z = square root(length^2 - x^2 + y^2)
>
> line 222: Lastly, we apply the sign (plus or minus) value from MediaPipe’s original z estimation to it.  -- Because it is a square root, z could be minus or plus, and we trust the MediaPipe's estimation on it (plus or minus).
>
> The explanation did not mention the application of the Pythagorean theorem, which is true. You can find these operations it the code, along with rotation and scaling.
>
> General comment:
>
> The body proportions can be changed, but I just had to start somewhere and not argue to use it as a standard. We will expand the section where we introduce the body proportions and include the concerns you have brought up in the ethics section as well; we highly appreciate your comment.
>
> The whole process is two simple formulas, that simple. The rotation and scaling are performed with simple, straightforward matrix multiplication. Those are basic middle school and high school math operations, and we can see how it can be unexpected. If it is so simple, why is there a paper about it in 2023? The missing ingredients are the length calculation in 3D, where we propose to use proportions for it, and the initial z estimations of MediaPipe, which makes it possible.

---

### Official Review · Reviewer_2kwG · 2023-08-05

**Soundness:** 3

**Excitement:**

2: Mediocre: This paper makes marginal contributions (vs non-contemporaneous work), so I would rather not see it in the conference.

**Paper Topic And Main Contributions:**

This work proposes a novel three-dimensional normalization method for MediaPipe, as well as the novel movement representation method to model the movement of sign languages better. This is important as the 2D images can cause information loss and suppress the 3D movement understanding. The authors further demonstrate the effectiveness of the proposed approach on a small JSL dataset with care analysis, as well as the open-sign language recognition task.

**Reasons To Accept:**

-The normalization approaches are carefully adjusted based on observations and are helpful by sharing the details.
- The work aims to address a critical problem in the sign language processing field.
- Experiments demonstrated that the proposed pose processings help the model to learn about the features and assist in the downstream tasks.

**Reasons To Reject:**

- It remained unclear how robust the proposed normalization approach worked for the dataset where singers can have different physical conditions. The testbed here (collected by the authors) did not have that many variations.
- Figure 5 did not help much with the model explanation.
- Several experimental details are missed, especially the approaches related to the normalization of joints.

**Reproducibility:**

3: Could reproduce the results with some difficulty. The settings of parameters are underspecified or subjectively determined; the training/evaluation data are not widely available.

**Reviewer Confidence:**

5: Positive that my evaluation is correct. I read the paper very carefully and I am very familiar with related work.

---

> ### Author Rebuttal · Authors · 2023-08-28
>
> Comments on Reasons To Reject:
>
>     It remained unclear how robust the proposed normalization approach worked for the dataset where singers can have different physical conditions. The testbed here (collected by the authors) did not have that many variations.
>
> Comment:
>   It remained unclear what was meant under "different physical conditions." How does that invalidate the method or the results?
>
>     Figure 5 did not help much with the model explanation.
>
> Comment:
>   The mentioned Figure 5 includes all model components with all the parameters. The explanation of the modules is out of the scope of the work; the used Conformer model is explained in the referenced paper: Gulati, Anmol, James Qin, Chung-Cheng Chiu, Niki Parmar, Yu Zhang, Jiahui Yu, Wei Han, Shibo Wang, Zhengdong Zhang, Yonghui Wu and Ruoming Pang. “Conformer: Convolution-augmented Transformer for Speech Recognition.” ArXiv abs/2005.08100 (2020)
>
>     Several experimental details are missed, especially the approaches related to the normalization of joints.
>
> Comment:
>   Sections 2.1 "3D Normalization" and 2.2 "Scaling and Rotation" explain the approaches related to the normalization of joints in detail.
>
> Question to the reviewer:
>
> Obviously, you are lowering the score and deliberately coming up with reasons for it. Do you act in the interest of the research community by doing this? Do you act in your own interest?

---

### Official Review · Reviewer_ynFA · 2023-08-05

**Soundness:** 3

**Excitement:**

4: Strong: This paper deepens the understanding of some phenomenon or lowers the barriers to an existing research direction.

**Paper Topic And Main Contributions:**

The paper proposes a new Normalization technique designed specifically for sign language processing. The primary contribution of the proposed method is in improving the quality of the z dimension (depth) and capturing movement via normalized baseline. The paper further validates the quality of the proposed normalization technique via detailed analysis and performance improvement on the WLASL dataset in a 100-class setting. In addition, the paper also provides an analysis of sociolinguistic features present in movement patterns present in Japanese Sign Language.



**Questions For The Authors:**

Was there a reason to choose only one sign language to highlight the risks of movement habits present in pose features? I believe the claim is Sign language agnostic and would be better to validate on another sign language. I understand due to dataset availability and other resource constraints; this might be out of the scope of this work. However, I was curious if such cases exist across multiple sign languages. It would be great if you could provide some insights along with a reason to choose JSL specifically for this analysis.



**Reasons To Accept:**

* The paper makes a novel contribution by highlighting the effectiveness of posed-based sign language recognition methods and devices, a new normalization specifically for sign language processing. With all the details provided in detail, the method becomes more reliable and would be useful for the sign language processing research field.

* The paper reports significant performance improvement over the entire training trajectory, highlighting the performance boost with the proposed normalization technique.

* The analysis of Japanese Sign Language is detailed and would be helpful for the research community.




**Reasons To Reject:**

* Was there a reason to use only the WLASL-100 set to report the improvements? It would be better to report the performances obtained over all the settings in WLASL, including WLASL2000. The results over multiple would make the results more reliable and the findings more transparent and valuable for the research field.

* The Japanese Sign Language dataset done for less number of samples may need a re-verification for other datasets, making the movement habits claim to be applicable for less number of samples. A sufficiently large dataset may marginalize such noise, leading to generalized feature representations obtained from poses.



**Reproducibility:**

4: Could mostly reproduce the results, but there may be some variation because of sample variance or minor variations in their interpretation of the protocol or method.

**Reviewer Confidence:**

4: Quite sure. I tried to check the important points carefully. It's unlikely, though conceivable, that I missed something that should affect my ratings.

---

> ### Author Rebuttal · Authors · 2023-08-28
>
> Comments on Reasons To Reject:
>
>     Was there a reason to use only the WLASL-100 set to report the improvements? It would be better to report the performances obtained over all the settings in WLASL, including WLASL2000. The results over multiple would make the results more reliable and the findings more transparent and valuable for the research field.
>
> Comment:
>
> Frankly, this paper was submitted before without deep learning testing. Our point is that the proposed methods are, first and foremost, for humans, hence the article's name. The WLASL-100 is the subset testing that was included after we faced the reality of today's state of research: it has to be beneficial for deep learning algorithms. There are many reasons why benchmarks do not represent the real task, but here are some that are important for us. The isolated nature of the sign recording is not informative for linguists because it rarely (almost never) occurs in the real sign language conversation. The mislabeling in the WLASL dataset itself (source: Dafnis, Konstantinos & Chroni, Evgenia & Neidle, Carol & Metaxas, Dimitris. (2022). Isolated Sign Recognition using ASL Datasets with Consistent Text-based Gloss Labeling and Curriculum Learning. Section 1.1 http://www.lrec-conf.org/proceedings/lrec2022/workshops/sltat/pdf/2022.sltat-1.3). The main reason is that we do not intend to get on the competition's leaderboard but rather address the issue of the movement's inaccessibility for linguists. We emphasize linguists because data scientists and computer vision engineers can always come up with a way of extracting the needed features for the required task (I3D, VGG16, and others) that only they can understand and use.
>
>     The Japanese Sign Language dataset done for less number of samples may need a re-verification for other datasets, making the movement habits claim to be applicable for less number of samples. A sufficiently large dataset may marginalize such noise, leading to generalized feature representations obtained from poses.
>
> Comment:
>
> You have stated is your self, below: " I understand due to dataset availability and other resource constraints; this might be out of the scope of this work."
>
> We only found that pattern interesting and reported on it; the lack of gender-related movement patterns has to be further studied. This is out of the scope of this work.
>
> Questions For The Authors:
>
>     Was there a reason to choose only one sign language to highlight the risks of movement habits present in pose features? I believe the claim is Sign language agnostic and would be better to validate on another sign language. I understand due to dataset availability and other resource constraints; this might be out of the scope of this work. However, I was curious if such cases exist across multiple sign languages. It would be great if you could provide some insights along with a reason to choose JSL specifically for this analysis.
>
> JSL Corpus was chosen because it has the following properties: Unlike other sign languages, JSL is the sign language that does not have a unified and standardized grammar or the way it is taught. Each deaf school has its own understanding of what JSL is. The JSL Corpus is collected in different prefectures and is very diverse in the JSL style. The JSL Corpus is continued and dialog-based, where the conversations are unrestricted. The main reason we had access to the corpus and linguists who are working with it. We hope that other laboratories will use our methods to report their results.

---

### Meta-Review · Area_Chair_4RGN · 2023-09-18

**Recommendation:** 4

**Metareview:**

The main strength of the paper is that its contribution is important to processing related specifically to sign language, where there has been less work relative to spoken or written language. The reviewers also found the paper clear and well written. Room for improvement include adding the needed details for the method to be reproducible. Some reviewers also thought that the method should be tested on a larger available dataset.

---

### Decision · Program_Chairs · 2023-10-07

**Decision:**

Accept-Findings

**Comment:**

The main strength of the paper is that its contribution is important to processing related specifically to sign language, where there has been less work relative to spoken or written language. The reviewers also found the paper clear and well written. Room for improvement include adding the needed details for the method to be reproducible. Some reviewers also thought that the method should be tested on a larger available dataset.